# STOCKBENCH: CAN LLM AGENTS TRADE STOCKS PROFITABLY IN REAL-WORLD MARKETS?

## ABSTRACT

Large language models (LLMs) have recently demonstrated strong capabilities as autonomous agents, showing promise in reasoning, tool use, and sequential decision-making. While prior benchmarks have evaluated LLM agents in domains such as software engineering and scientific discovery, the finance domain remains underexplored, despite its direct relevance to economic value and high-stakes decision-making. Existing financial benchmarks primarily test static knowledge through question answering, but they fall short of capturing the dynamic and iterative nature of trading. To address this gap, we introduce STOCKBENCH, a contamination-free benchmark designed to evaluate LLM agents in realistic, multi-month stock trading environments. Agents receive daily market signals—including prices, fundamentals, and news—and must make sequential buy, sell, or hold decisions. Performance is assessed using financial metrics such as cumulative return, maximum drawdown, and the Sortino ratio. Our evaluation of state-of-the-art proprietary (*e.g.,* GPT-5, Claude-4) and open-weight (*e.g.,* Qwen3, Kimi-K2, GLM-4.5) models shows that while most LLM agents struggle to outperform the simple buy-and-hold baseline, several models demonstrate the potential to deliver higher returns and manage risk more effectively. These findings highlight both the challenges and opportunities in developing LLM-powered financial agents, showing that excelling at static financial knowledge tasks does not necessarily translate into successful trading strategies. We release STOCKBENCH as an open-source resource to support reproducibility and advance future research in this domain.

## 1 INTRODUCTION

Large language models (LLMs) have enabled a new wave of autonomous agents, demonstrating strong capabilities in reasoning, tool use, and long-horizon decision making (OpenAI, 2024; Anthropic, 2025a; DeepMind, 2025; Liu et al., 2024; Guo et al., 2025a; Meta-AI, 2025; Yang et al., 2024a; Bai et al., 2025; OpenAI, 2025b). This agentic capability is verified by benchmarks in various different domains, such as software engineering (Jimenez et al., 2024; Yang et al., 2024b), scientific discovery (Mialon et al., 2023), and marketing (Chen et al., 2025; Barres et al., 2025), using the most recent advanced LLMs such as GPT-5 (OpenAI, 2025a) and Claude-4 (Anthropic, 2025b), highlighting their promise for workflow automation and productivity gains. The ever-evolving agent capability of LLMs pushes agent application toward real-world productivity and economic value.

Among various agent application scenarios, the finance domain stands out due to its direct connection to economic value and the high stakes involved in decision making (Wu et al., 2023; Lee et al., 2024; Nie et al., 2024). To holistically evaluate the profitability and risk-management capabilities of LLM agents in finance, an ideal benchmark should adhere to three key principles: **(1) Realistic Market Interaction.** The agent must operate in a dynamic market environment, responding to real-time price movements and news events. **(2) Continuous Decision Making.** The agent should make sequential trading decisions over an extended horizon, reflecting the iterative nature of investment strategies. **(3) Data Contamination Free.** To ensure fair evaluation, the agent must not have prior exposure to the test data during training, necessitating careful data curation and temporal separation.

However, existing benchmarks for financial agents largely focus on static question-answering tasks (Chen et al., 2021; Zhu et al., 2021; Yin et al., 2023), which are designed to test the financial knowledge coverage of LLMs but fail to reflect practical trading scenarios. Although recent

Table 1: Comparison of STOCKBENCH with existing financial benchmarks.

| Benchmark | Market Simulation | Multi Month Horizon | Continuous Decision | Contamination Free | Direct Economic Value |
|---|---|---|---|---|---|
| FinQA (Chen et al., 2021) | ✗ | ✗ | ✗ | ✗ | ✗ |
| ConvFinQA (Chen et al., 2022) | ✗ | ✗ | ✗ | ✗ | ✗ |
| FLUE (Shah et al., 2022) | ✗ | ✗ | ✗ | ✗ | ✗ |
| FinEval (Guo et al., 2025b) | ✗ | ✗ | ✗ | ✗ | ✗ |
| CPA-QKA (Kuang et al., 2025) | ✗ | ✗ | ✗ | ✗ | ✗ |
| BizFinBench (Lu et al., 2025) | ✗ | ✗ | ✗ | ✗ | ✗ |
| Finance Agent Benchmark (Bigeard et al., 2025) | ✓ | ✗ | ✓ | ✗ | ✗ |
| INVESTORBENCH (Li et al., 2024) | ✓ | ✓ | ✓ | ✗ | ✓ |
| FinSearchComp (Hu et al., 2025) | ✗ | ✓ | ✓ | ✗ | ✓ |
| **STOCKBENCH (Ours)** | ✓ | ✓ | ✓ | ✓ | ✓ |

efforts like INVESTORBENCH (Li et al., 2025a) take a step towards simulating trading environments, this thread of works only focuses only on single-stock-trading and is conducted on historical data prior to 2021, raising concerns about potential data contamination.

To mitigate the gap, we propose STOCKBENCH, an evolving benchmark that places LLM agents into realistic stock-trading environments, directly measuring their profitability and risk-management capabilities. Specifically, STOCKBENCH is designed to be: **(1) Realistic.** Agents receive daily market signals including prices, company fundamentals, and news headlines, reflecting real-world trading contexts. **(2) Continuous.** Agents must make sequential daily trading decisions (buy, sell, or hold) over a multi-month horizon, mirroring the iterative nature of investment strategies. **(3) Contamination-Free.** The benchmark is instantiated using recent market data from March 2025 to July 2025 and will be continuously updated to avoid overlap with the training corpora of contemporary LLMs. Performance is evaluated using key financial metrics such as cumulative return, maximum drawdown, and the Sortino ratio, providing a direct and quantitative assessment of trading success.

As a proof of concept, we evaluate a diverse set of LLM agents, including both proprietary models (*e.g.,* GPT-5 (OpenAI, 2025a), Claude-4 (Anthropic, 2025b)) and open-weight models (*e.g.,* Qwen3 (Yang et al., 2025), Kimi-K2 (Team et al., 2025), GLM-4.5 (Zeng et al., 2025)), alongside an equal-weight buy-and-hold baseline. Surprisingly, despite their strong performance on financial QA benchmarks, most LLM agents fail to outperform this simple baseline in terms of both cumulative return and risk-adjusted return. This finding suggests that excelling at static QA does not necessarily translate into effective trading strategies in dynamic market environments, underscoring a key challenge in the development of LLM-powered financial agents.

The main contributions of this work are summarized as follows:

- We introduce STOCKBENCH, a novel benchmark for evaluating LLM agents in realistic stock-trading environment, directly measuring their profitability and risk-management capabilities.

- We design a comprehensive evaluation framework that incorporates realistic market dynamics, diverse input data, and multiple financial metrics to holistically assess agent performance.

- We conduct extensive experiments by implementing various backbone LLM as stock-trading agents, revealing their current limitations in achieving profitable trading strategies and underscoring the need for further advancements in this domain.

- We open-source implementation of STOCKBENCH to facilitate reproducibility and encourage community contributions, fostering further research on LLM-powered financial agents.

## 2 STOCKBENCH

The construction of STOCKBENCH consists of two main building blocks. (1) A back-trading environment, which contains historical data necessary for stock-trading decision making. We simulate real-world stock trading using this back-trading setup. (2) An associated stock-trading agent workflow. This workflow allows us to evaluate LLM backbones as agents to engage in the back-trading environment. The overall framework of STOCKBENCH is demonstrated in Figure 1.

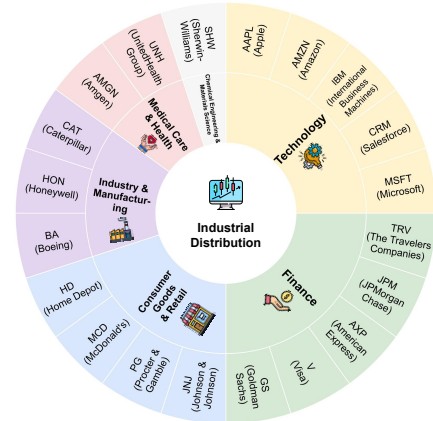

Figure 1: Overview of STOCKBENCH. The design of STOCKBENCH includes a back-trading benchmark dataset, and an associated workflow that converts backbone LLMs into agents.

## 2.1 BACK-TRADING ENVIRONMENT

We design the back-trading environment to simulate realistic stock trading, where trading agents are exposed only to data available up to the time of each decision. To set up the environment, we identify three critical sources of information for trading decision making: (1) A bundle of investment targets, which defines the scope of the environment. We pre-define these investment targets to facilitate reproducibility of the evaluation on STOCKBENCH. (2) Historical market data, which includes both the prices and fundamental indicators. These enable the evaluated trading agents to perform quantitative analysis. (3) News corpora, which capture events that drive stock price fluctuations. We elaborate on the data collection process below.

**Investment Targets.** The investment targets are a bundle of stocks that allow the trading agents to perform buy and sell operations. We manually select the investment targets in STOCKBENCH to prevent potential outcome fluctuations caused by stock selection—*e.g.,* trading agents might otherwise happen to pick a stock driven by irrational market sentiment—thereby stabilizing the evaluation results.

To this end, we select 20 stocks from the Dow Jones Industrial Average (DJIA) with the highest weights as our investment targets. In particular, high-weighted DJIA stocks are representative of the global stock market and are less prone to short-term irrational sentiment-driven events. Constraining the trading action space to our selected investment targets mirrors real-world investor attention while keeping the dataset computationally tractable. Moreover, information about these well-known stocks is transparent and easy to collect, being readily accessible through web search engines. We show the distribution of the selected investment targets across different industries in Figure 2. Our selection covers technology, finance, and manufacturing, ensuring stock diversity.

Figure 2: Industry distribution of selected stocks.

**Historical Market Data.** We collect and preserve historical market data containing key quantitative information. For each stock, we use official opening prices together with a concise set of fundamental metrics such as market capitalization, price-to-earnings (P/E) ratio, dividend yield, and trading range. These signals provide a reliable snapshot of company health and valuation, supporting informed decision making. We also retain the timestamps of the collected data to prevent any leakage of future information to the agent.

**News Corpora.** We construct news corpora for stocks to enable stock-trading agents to interpret both sentiments and events in a manner that resembles how retail investors react to market narratives. For each stock, we collect news articles released within the previous 48 hours on a daily basis. These articles are retrieved using news-search API[1] with time restrictions. Since news analysis consumes substantial context length in backbone LLMs, we balance information coverage and computational cost by preserving the top five relevant news articles each time the search engine returns results.

We also carefully select the time window for collecting data in the back-trading environment. In principle, the evaluation window should satisfy two conditions: (1) the included stock information must not have been exposed to the evaluated stock-trading agents during their model training stages; and (2) the window should be sufficiently long to mitigate the impact of random noise that affects only short periods of time. To this end, we collect data spanning from `March 3, 2025` to `June 30, 2025`, a four-month period that includes both volatility and trend reversals. This period also falls after the knowledge cutoff of mainstream LLMs, ensuring no data leakage. It is worth noting that we will continuously update the back-trading environment to avoid overlap with the training corpora of contemporary LLMs.

## 2.2 STOCK-TRADING AGENT WORKFLOW

We provide a stock-trading agent workflow that enables backbone LLMs to interact with the back-trading environment as agents. The design of the workflow follows two goals. (1) Minimal workflow. We keep the workflow minimal, since overly complicated workflows introduce inductive biases that may favor certain backbone LLMs. (2) Realistic. We design the workflow to align with the iterative decision-making process of retail investors.

In particular, we follow previous frameworks (Zhang et al., 2020; Tsantekidis et al., 2017; Moody & Saffell, 2001; Deng et al., 2016) and organize the stock-trading workflow into four essential stages: portfolio overview, in-depth stock analysis, decision generation, and execution and validation. Overall, the design prioritizes realism, fairness, and reproducibility, in line with earlier studies on benchmark construction for trading environments.

**Step 1: Portfolio Overview.** The agent first scans all available stocks in the market (the "investment target"), receiving relevant data for each stock. This includes recent news, current holdings of the agent, historical actions, and the opening price. This step mirrors how a trader assesses the broader market and the overall status of each stock in their portfolio.

**Step 2: In-Depth Stock Analysis.** After the initial overview, the agent selects specific stocks for deeper analysis. For these selected stocks, the agent is provided with additional fundamental data such as market capitalization, P/E ratio, and dividend yield. This step simulates how a trader focuses on a subset of stocks identified in the initial overview, examining their financial health and other key metrics in greater depth.

**Step 3: Decision Generation.** With the enriched context, the agent generates decisions for each stock, choosing between three possible actions: (1) increase, (2) decrease, or (3) hold the position. These options ensure that actions of the agent are clear, actionable, and executable within the constraints of a retail investor's decision making process.

**Step 4: Execution and Validation.** Finally, the decisions are executed by converting dollar targets into share quantities based on the opening price. If the decisions of the agents exceed available liquidity, the system flags the issue and requires the agent to revise its decisions until they can be executed within available resources. Once validated, the new portfolio weights are locked, and the simulation advances to the next day.

## 2.3 FEATURES OF STOCKBENCH

We now discuss how the design of STOCKBENCH satisfies the following key principles:

**Realistic Market Interaction.** The design of the back-trading environment mimics real-world trading scenarios through three key elements: (1) a carefully selected bundle of investment targets, (2) reliable price and fundamental data, and (3) a concise yet timely news corpus. These elements en-

---

[1] https://finnhub.io/

sure that the agent is exposed to information mirroring the complexities of real trading environments, while avoiding unrealistic or overly expansive inputs.

**Continuous Decision Making.** In the workflow, the agent first performs a portfolio overview, then conducts in-depth stock analysis, and finally generates daily trading decisions (buy, sell, or hold) based on this analysis. These steps reflect the continuous decision-making process of retail investors, enabling the agent to adapt its strategies over time in response to market conditions.

**Data Contamination Free.** We ensure that the agent has no prior exposure to the test data during its training. To achieve this, the benchmark is instantiated using recent market data, ensuring temporal separation and avoiding any overlap with the training corpora of contemporary LLMs.

## 3 MAIN EXPERIMENTS

In this section, we present the experimental setup and results of evaluating various LLM agents within the STOCKBENCH trading workflow. We describe the trading environment, selected models, baseline strategy, and evaluation metrics. We then analyze performance outcomes, highlighting key insights into the capabilities of LLM agents in real-world financial markets.

### 3.1 EXPERIMENT SETUP

We detail the experimental setup for evaluating LLM agents in the STOCKBENCH trading workflow. Specifically, we describe the trading environment, the models selected for benchmarking, the passive baseline, and the evaluation metrics used to assess performance.

**Trading Environment.** The top 20 DJIA stocks are selected as the investment targets, ensuring diverse representation across sectors. The evaluation period spans four months, from March 3 to June 30, 2025, covering 82 trading days and capturing a range of market conditions. Each model starts with $100,000$ in cash and zero holdings, making daily trading decisions at market open. Key inputs include (1) the historical actions on held stocks over the past seven days, (2) up to five recent news articles from the previous 48 hours, and (3) for selected stocks, fundamental data such as market capitalization, P/E ratio, dividend yield, 52-week high/low, and recent quarterly dividends.

**Models to Evaluate.** We benchmark a diverse set of LLMs, including both open-weight models such as Qwen3 (Yang et al., 2025)[2], DeepSeek (Guo et al., 2025a; Liu et al., 2024), Kimi-K2 (Team et al., 2025), GLM-4.5 (Zeng et al., 2025) and GPT-OSS (OpenAI, 2024), as well as closed-source APIs like OpenAI's O3 (OpenAI, 2025b) and Anthropic's Claude-4-Sonnet (Anthropic, 2025b). This selection covers a range of architectures, sizes, and training methodologies to assess generality across different LLM designs. All models are equipped with $32,768$ token context windows and decoded with official recommended settings to ensure their performance is optimized for the task. To hance a reliable result, each LLM agents would be run three times with different random seeds, and the average performance is reported.

**Passive Baseline.** As a reference point, we implement a passive equal-weight buy-and-hold strategy that allocates the initial capital equally across all selected stocks at the start of the evaluation period and holds these positions unchanged until the end. This naive allocation is a widely accepted benchmark in portfolio research, reflecting passive index tracking behavior and providing a robust lower bound against which more sophisticated active strategies can be compared (DeMiguel et al., 2009; Duchin & Levy, 2009).

**Evaluation Metrics.** We adopt three widely used measures in financial analysis:

*Final Return.* This metric captures overall profitability as the percentage change in portfolio value from the initial amount $V_0$ to the final amount $V_T$:

$$\text{Final Return} = \frac{V_T - V_0}{V_0} \qquad (1)$$

It directly reflects the portfolio's overall performance over the evaluation period and is a simple, widely used measure of investment profitability (Bodie et al., 2014).

---

[2]Without special denote, the Qwen3 series in this papers refers to the 2507 variants

*Maximum Drawdown.* The maximum drawdown quantifies the largest decline in portfolio value from its peak to its trough during the evaluation period, providing a measure of downside risk:

$$\text{Max Drawdown} = \min_{t \in [0,T]} \left( \frac{V_t - \max_{s \leq t} V_s}{\max_{s \leq t} V_s} \right) \tag{2}$$

It highlights the worst loss an investor could have faced and is commonly used to assess risk and volatility (Magdon-Ismail et al., 2004; Chekhlov et al., 2005).

*Sortino Ratio.* The Sortino ratio is a risk adjusted return metric that penalizes only downside volatility. It is defined as the excess return $R_p$ divided by the downside deviation $\sigma_d$:

$$\text{Sortino Ratio} = \frac{R_p}{\sigma_d}, \quad \sigma_d = \sqrt{\frac{1}{N_d} \sum_{i=1}^{N_d} \min(R_i, 0)^2} \tag{3}$$

This metric is more appropriate than the Sharpe ratio when returns are asymmetric, as it focuses on negative volatility (Sortino & Van der Meer, 1991; Pedersen & Satchell, 2002).

After computing these metrics for each model, we derive a composite rank by leveraging the z-score of each metric, averaging them to produce a single performance score.

$$\text{Composite Rank} = \frac{z(\text{Final Return}) - z(\text{Max Drawdown}) + z(\text{Sortino Ratio})}{3} \tag{4}$$

This approach balances profitability and risk, rewarding models that achieve high returns while effectively managing downside exposure.

## 3.2 EXPERIMENT RESULTS

Table 2 presents the performance of all evaluated models over the four-month period without contamination. The results are reported across three key metrics—percentage return, maximum drawdown, and Sortino ratio—along with an overall ranking derived from a composite z-score of these metrics.

Here are the key observations: **(1) LLM agents can trade profitably in real-world markets.** Most tested models outperform the passive buy-and-hold baseline, which achieves a modest $0.4\%$ return with a $-15.2\%$ drawdown and a Sortino ratio of $0.0155$. Several agents deliver returns above $2\%$, with improved risk profiles. **(2) LLM agents can manage downside risk effectively.** All tested models achieve lower maximum drawdowns than the baseline, indicating that they can mitigate losses during market downturns. The best-performing agents limit drawdowns to around $-11\%$ to $-14\%$, compared to the baseline's $-15.2\%$. **(3) Reasoning model does not guarantee better performance.** Although reasoning-tuned models such as Qwen3-235B-Think and Qwen3-30B-Think exhibit strong performance in tasks requiring complex reasoning, including math and coding (Yang et al., 2025), they do not consistently outperform instruction-tuned counterparts

Table 2: The performance of tested models over the evaluation period. The best performance in each metric is highlighted in bold. Models are ranked based on the z-score aggregation of all three metrics. RT stands for Final Return (%), DDN stands for Max Drawdown (%).

| Model | RT | DDN | Sortino | Rank |
|---|---|---|---|---|
| Kimi-K2 | 1.9 | $-11.8$ | **0.0420** | 1 |
| Qwen3-235B-Ins | 2.4 | $\mathbf{-11.2}$ | 0.0299 | 2 |
| GLM-4.5 | 2.3 | $-13.7$ | 0.0295 | 3 |
| Qwen3-235B-Think | **2.5** | $-14.9$ | 0.0309 | 4 |
| OpenAI-O3 | 1.9 | $-13.2$ | 0.0267 | 5 |
| Qwen3-30B-Think | 2.1 | $-13.5$ | 0.0255 | 6 |
| Claude-4-Sonnet | 2.2 | $-14.2$ | 0.0245 | 7 |
| DeepSeek-V3.1 | 1.1 | $-14.1$ | 0.0210 | 8 |
| GPT-5 | 0.3 | $-13.1$ | 0.0132 | 9 |
| Qwen3-Coder | 0.2 | $-13.9$ | 0.0137 | 10 |
| DeepSeek-V3 | 0.2 | $-14.1$ | 0.0144 | 11 |
| Passive Baseline | 0.4 | $-15.2$ | 0.0155 | 12 |
| GPT-OSS-120B | $-0.9$ | $-14.0$ | 0.0156 | 13 |
| GPT-OSS-20B | $-2.8$ | $-14.4$ | $-0.0069$ | 14 |

Table 3: Performance of representative models (Kimi-K2 and GPT-OSS-120B) across different investment target sizes. Results are reported as mean return (% Mean), standard deviation of returns (% Std), and coefficient of variation (CV).

| Stocks | % Mean | % Std | CV |
|---|---|---|---|
| *Kimi-K2* | | | |
| 5 | $-4.6$ | 0.7 | 0.2 |
| 10 | 3.2 | 0.6 | 0.2 |
| 20 | 1.9 | 1.7 | 0.9 |
| 30 | $-0.5$ | 1.2 | 2.2 |
| *GPT-OSS-120B* | | | |
| 5 | $-5.7$ | 0.3 | 0.1 |
| 10 | 2.5 | 0.4 | 0.2 |
| 20 | $-0.4$ | 3.9 | 10.2 |
| 30 | $-0.9$ | 3.9 | 4.4 |

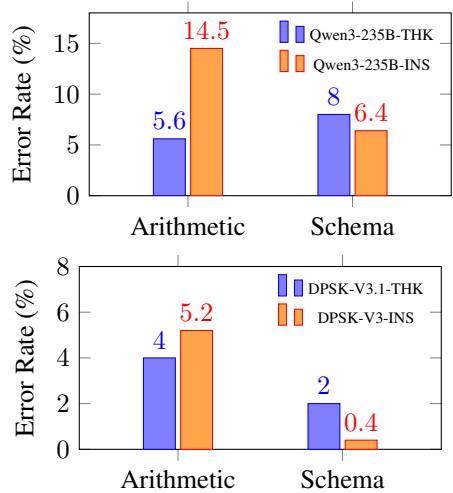

Figure 3: Error distribution (%) by type for Think vs Instruct models.

in this trading task. For example, Qwen3-235B-Ins outperforms its reasoning-tuned version with a lower maximum drawdown ($-11.2\%$ vs. $-14.9\%$). This suggests there is still a gap between reasoning ability and effective decision-making in dynamic, noisy environments like financial markets.

## 4 ANALYSIS

### 4.1 THE INFLUENCE OF INVESTMENT TARGET SIZE

To evaluate the impact of the investment target size on the agent's performance, we conducted the daily trading task with investment targets of 5, 10, 20, and 30 DJIA constituents, repeating the task three times and recording portfolio-weight differences across runs. The results show that variability increases as the investment target expands.

Specifically, as shown in Table 3, **(1) Scalability is inherently challenging.** All evaluated models exhibit performance degradation as the investment portfolio size increases, characterized by declining mean returns and rising return volatility. This indicates that scaling the number of tradable assets poses a non-trivial challenge for LLM agents. **(2) Model scale confers robustness.** The larger-scale model, Kimi-K2, demonstrates greater robustness to portfolio expansion, maintaining relatively stable risk-return profiles and achieving positive expected returns at moderate portfolio sizes (e.g., 10–20 stocks), whereas the smaller GPT-OSS-120B suffers from severe performance deterioration and excessive variability, suggesting that increased model capacity enhances generalization and stability in multi-asset decision-making contexts.

### 4.2 THE INFLUENCE OF ERROR IN THE TRADING WORKFLOW

During the trading process, various error happened during the agent's interaction with the environment. The most two common errors are: **(1) Arithmetic Error**, where the agent makes mistakes in calculating the number of shares to buy or sell based on the provided budget and stock price. **(2) Schema Error**, where the agent fails to adhere to the specified JSON output format, leading to parsing failures.

Figure 3 illustrates the frequency of these errors across thinking models and instruct models. Specifically, we observe that: Thinking models demonstrate a lower incidence of arithmetic errors compared to instruct models, this observation aligns that thinking models' outstanding performance in reasoning tasks such as math reasoning (Yu et al., 2025; Guo et al., 2025a; Yang et al., 2025). However, as for schema errors, thinking models exhibit a higher frequency of such errors compared to instruct models. This discrepancy aligns with recent findings that reasoning model tend to overthink

and produce more complex outputs, which can lead to deviations from the expected format (Fu et al., 2025; Li et al., 2025b).

### 4.3 ABLATION STUDY ON DATA SOURCES

In our workflow, LLM agents rely primarily on two types of information sources: news articles and fundamental financial data. These two modalities provide complementary signals, with news capturing market sentiment and fundamentals grounding the model in key financial indicators. To better understand their respective contributions, we conduct an ablation study by progressively removing these inputs.

Table 4: The cumulative return (CR, %) for Kimi-K2 and GPT-OSS-120B under three input settings: full input (Full), without news articles (w/o News), and without both news and fundamental data (w/o News & Fund.).

| Condition | Return (%) |
|---|---|
| *Kimi-K2* | 1.9 |
| *w/o* News | 1.4 |
| *w/o* News & Fund. | 0.6 |
| *GPT-OSS-120B* | −1.2 |
| *w/o* News | −1.2 |
| *w/o* News & Fund. | −3.4 |

As shown in Table 4, the cumulative return decreases consistently as we remove news and then fundamental data. This behavior matches our expectation that both information sources play an important role in guiding trading decisions. The Kimi-K2 model remains relatively robust when only news is removed, but its performance deteriorates when both inputs are absent. In contrast, GPT-OSS-120B experiences a sharper decline, indicating that it relies more heavily on explicit signals provided by news and fundamentals. Overall, these findings highlight that LLM-based trading agents are capable of integrating heterogeneous inputs, combining textual information from news with numerical fundamentals to produce more informed and effective trading strategies.

### 4.4 IMPACT OF EVALUATION WINDOW

A good trading model should be able to adapt to changing market conditions over time. To investigate how the choice of evaluation window affects model rankings, we conduct experiments using two different time frames: a downturn period (January to April 2025) and a upturn period (May to August 2025) with Kimi-K2, DeepSeek-series model, GPT-OSS series model and the passive baseline as references. Through this analysis, we aim to understand how models perform under different market regimes and whether their profitability and risk profiles shift accordingly.

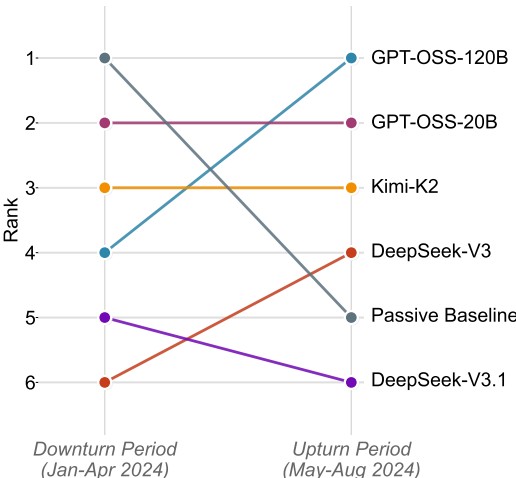

Figure 4: Model performance ranking based on the cumulative return, over two evaluation windows downturn (Jan-Apr 2025) and upturn (May-Aug 2025).

Figure 4 presents the ranking of models based on cumulative return across the two evaluation windows. Notably, we observe significant shifts in model rankings between the downturn and upturn periods. For instance, GPT-OSS-120B, which ranks shift from the bottom during the downturn to the top during the upturn, indicating that it may be better suited to bullish market conditions. While Kimi-K2 maintains a relatively stable ranking across both periods, suggesting its robustness to market fluctuations. This suggests that certain models may be better suited to specific market conditions, potentially due to their underlying architectures or training data. Besides, we also observe that during the downturn period, all the LLM agents failed to outperform the passive baseline, while in the upturn period, most LLM agents surpass the baseline. This indicates that LLM agents may struggle to navigate bearish markets, highlighting a key area for future improvement.

## 5 RELATED WORK

### 5.1 LLM AGENTS AND GENERAL BENCHMARKS

Large language models (LLMs) have rapidly progressed from powerful text completion systems to autonomous agents capable of reasoning, planning, and interacting with external environments (OpenAI, 2024; Anthropic, 2025a; DeepMind, 2025; Liu et al., 2024; Guo et al., 2025a; Meta-AI, 2025; Yang et al., 2024a; Bai et al., 2025; OpenAI, 2025b). There is growing consensus that agentic behavior represents the next stage of LLM development, as it directly connects language understanding with real-world productivity and economic value (OpenAI, 2025a; Anthropic, 2025b). In this paradigm, LLMs are not only evaluated on their static knowledge but also on their ability to continuously perceive, decide, and act.

To capture these emerging capabilities, a variety of benchmarks have been introduced across domains. For example, SWE-Bench (Jimenez et al., 2024) and SWE-Agent (Yang et al., 2024b) target software engineering tasks, GAIA (Mialon et al., 2023) focuses on scientific discovery, and marketing-oriented benchmarks such as XBench (Chen et al., 2025) and Tau2Bench (Barres et al., 2025) examine commercial workflows. These benchmarks highlight the promise of LLM agents for complex, multi-step problem solving and workflow automation. However, despite their breadth, few existing efforts have examined domains where decision-making is directly tied to measurable economic outcomes, such as financial trading.

### 5.2 FINANCIAL AGENTS AND BENCHMARKS

The financial domain has long been of interest for LLM applications due to its direct link with profitability, risk management, and high-stakes decision making (Wu et al., 2023; Lee et al., 2024; Nie et al., 2024). Most existing benchmarks, however, focus on static question-answering tasks such as FinQA (Chen et al., 2021), TAT-QA (Zhu et al., 2021), and FinBench (Yin et al., 2023). While useful for evaluating financial reasoning and domain knowledge, these tasks do not reflect the iterative, dynamic nature of real-world trading environments.

Recent work has begun to move towards more realistic evaluation settings. For instance, IN-VESTORBENCH (Li et al., 2025a) introduces an environment for testing trading decisions, marking an important step towards agent-based financial evaluation. However, it primarily considers single-stock-trading and relies on historical data up to 2021, raising concerns about both scope and potential data contamination.

In contrast, our proposed benchmark, STOCKBENCH, is the first to embed LLM agents into realistic, multi-stock-trading environments with continuously updated market data. By requiring agents to make sequential trading decisions over extended horizons, STOCKBENCH directly evaluates profitability and risk management capabilities. This setting bridges the gap between static financial QA benchmarks and the practical challenges of real-world investment strategies, enabling a more faithful assessment of the readiness of LLM-powered financial agents.

## 6 CONCLUSION

In this work, we introduce STOCKBENCH, a novel benchmark designed to evaluate the performance of LLM agents in realistic stock-trading scenarios. By simulating dynamic market environments and requiring continuous decision-making over multi-month horizons, STOCKBENCH provides a comprehensive framework to assess both profitability and risk management capabilities. Our extensive experiments reveal that while current LLM agents could operate profitably, they still struggle to consistently outperform simple baselines, highlighting the challenges that remain in this domain. We believe that STOCKBENCH will serve as a valuable resource for the research community, driving further advancements in the development of intelligent, autonomous financial agents capable of navigating complex market dynamics. Future work will focus on enhancing the benchmark with additional market scenarios and exploring novel agent architectures to improve trading performance.

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

## A  ETHICAL STATEMENT

We strictly comply with all applicable financial regulations, data-protection laws, and academic ethical standards during the construction and use of StockBench. All market data (prices, fundamentals, and news) were collected through licensed data vendors or public APIs that explicitly allow research use; no non-public, insider, or personally identifiable information was accessed or stored. The benchmark is provided for academic and non-commercial research purposes only. Users are reminded that StockBench is not intended to offer, or serve as the basis for, any financial advice, trading recommendation, or commercial activity. Any trading strategy tested on StockBench carries inherent market risk; past performance recorded in the benchmark does not guarantee future returns.

## B  REPRODUCIBILITY STATEMENT

To ensure the reproducibility of our work and foster further research in this domain, we plan to open-source our StockBench benchmark, including the dataset and the code for the back-trading workflow. This release will enable other researchers to replicate our experiments, validate our findings, and build upon our methodology.

## C  PREVENT DATA LEAKAGE

In this study, we minimize the risk of data leakage by carefully planning and evaluating the time frame. When testing large language models (LLMS) in the financial field, a potential concern is that during the training process, the model will learn a lot of past financial knowledge, which may lead to the model's performance being artificially exaggerated. For instance, when asking GPT-5 (without using the search function), we found that the model could accurately predict the stock trend of AAPL in 2021, and the model's response was consistent with the facts.

This discovery indicates that if the evaluation time is relatively early, the model may have obtained future information that could not have been reasonably acquired at the time of evaluation. In view of this, we have decided to limit the data used for evaluation to a more recent time frame, thereby minimizing the possibility of such "data leakage" and ensuring that the model is tested more fairly. By focusing on a narrow evaluation time window, we aim to simulate real-world scenarios where agents can only make trading decisions based on the publicly available information at the time of each decision.

This approach conforms to the best practices of financial model evaluation, ensuring that the evaluation results truly reflect the predictive and decision-making capabilities of LLM agents without being disturbed by the unintentional availability of future data

Table 5: Model Return Variance Across Different Models. This table presents the variance of model returns for various LLMs.

| Rank | Model | Var ($\times 10^{-4}$) |
|---|---|---|
| 1 | *DeepSeek-V3* | 0.074 |
| 2 | *DeepSeek-V3.1* | 0.203 |
| 3 | *GPT-5* | 0.210 |
| 4 | *Claude-4-Sonnet* | 0.153 |
| 5 | *GLM-4.5* | 0.099 |
| 6 | *Qwen3-30B-Think* | 0.115 |
| 7 | *Qwen3-235B-Think* | 0.321 |
| 8 | *Qwen3-235B-Ins* | 0.281 |
| 9 | *Qwen3-4B-Ins* | 1.382 |
| 10 | *GPT-OSS-20B* | 1.337 |
| 11 | *Qwen3-Coder* | 1.655 |
| 12 | *Openai-O3* | 3.250 |
| 13 | *Kimi-K2* | 1.866 |
| 14 | *GPT-OSS-120B* | 10.19 |

# D   MODEL RETURN VARIANCE

In this section, we analyze the return variances of different models. Models with higher return variances may exhibit more unpredictable behaviors, which is undesirable in many real-world applications, especially in high-risk environments such as financial decision-making.

We ranked several large language models (LLMS) based on their return variances, as shown in table 5. In the evaluated model, *DeepSeek-V3* exhibited the smallest performance fluctuation, indicating high stability. In contrast, *GPT-OSS-120B* exhibits the highest return variance, indicating a volatility in its performance.

# E   THE USE OF LARGE LANGUAGE MODELS

We use LLMs for two purposes. (1) Code implementation. When implementing the code for this paper, including data gathering and experiment implementation, we use LLMs in the form of `copilot` to complete code snippets. The architecture design is conducted by human researchers. (2) Proofreading. To fix grammar issues, we use LLMs as a writing tools to refine the draft.

We would like to highlight that LLMs are not responsible for creativity tasks during conducting the research of this paper, including but not limited to: ideation, experiment design, paper organizing.

