# OpenReview forum: "StockBench:Can Llm Agents Trade Stocks Profitably In Real-world Markets?"
_ICLR.cc/2026/Conference — ICLR 2026 Conference Withdrawn Submission_

### Official Review · Reviewer_XKW2 · 2025-10-15

**Soundness:** 1
**Presentation:** 2
**Contribution:** 2
**Rating:** 2
**Confidence:** 4

**Summary:**

The paper introduces STOCKBENCH, a benchmark designed to evaluate large language model (LLM) agents in realistic multi-month stock-trading environments. It simulates daily trading over a four-month window (Mar–Jun 2025) using 20 Dow Jones constituents, with inputs including fundamentals, prices, and news. Models (GPT-5, Claude-4, Qwen3, Kimi-K2, etc.) are compared against a buy-and-hold baseline using financial metrics like cumulative return, maximum drawdown, and Sortino ratio. Results show that most LLM agents fail to outperform the baseline, though some demonstrate improved risk control. The paper claims this benchmark is “contamination-free” and open-sourced to facilitate reproducibility.

**Strengths:**

- Targets a meaningful gap between static financial QA and dynamic trading evaluation.

- Potentially interesting results that reasoning-tuned models (e.g., Qwen-Think) do not outperform simpler instruction-tuned versions in practice.

**Weaknesses:**

- The evaluation period is only four months, which is clearly insufficient for daily trading evaluation. While it is understandable that the authors wish to avoid data contamination given LLMs’ knowledge cut-off, a four-month window is too short to draw any statistically meaningful conclusions. I would prefer to see a longer evaluation period, even if it introduces potential look-ahead bias. In fact, existing literature has already shown that when tested over longer horizons of around twenty years, LLMs still fail to outperform the market even under potential data leakage, which would provide a more credible baseline.

- Although the authors claim to evaluate “LLM agents”, key prior works such as FinMem and FinAgent are not mentioned, compared, or benchmarked. Without referencing or contrasting these existing frameworks, the claim of evaluating agentic capabilities is unconvincing. Moreover, the proposed demo framework does not show any workflow of tool usage or memory management, which are essential components of agentic design. The entire description remains highly abstract and lacks sufficient technical details to substantiate its claimed novelty.

- The simulation design itself appears incomplete and not properly implemented. There is no consideration of basic trading realism, including commission fees, slippage, or liquidity caps, all of which are crucial for evaluating profitability in real markets. Ignoring these factors leads to overly optimistic and practically meaningless results, weakening the validity of the evaluation.

- The results are presented without any statistical robustness checks. There is no analysis of significance, confidence intervals, or variance decomposition to determine whether observed performance differences are meaningful rather than random fluctuations. Moreover, LLM outputs are affected by sampling randomness such as temperature, but these sources of uncertainty are not quantified or controlled.

There are some other minor issues, such as clarity, technical details, and further analysis. However, I believe the weaknesses mentioned above are the most critical, as the claims are not well-supported, and the entire evaluation itself might potentially be problematic.

**Questions:**

- Can the authors provide concrete examples of the agent’s input, reasoning trace, and action output to clarify how decisions are made?

- What's the temperature setting?

- Are differences in returns statistically significant, or within the range of random noise? What's the alpha and beta?

---

### Official Review · Reviewer_5AJA · 2025-10-29

**Soundness:** 2
**Presentation:** 3
**Contribution:** 2
**Rating:** 4
**Confidence:** 4

**Summary:**

This paper introduces STOCKBENCH, a contamination-free benchmark for evaluating LLM agents in realistic multi-month stock trading simulations. Performance is measured using standard financial metrics (e.g., cumulative return, maximum drawdown, Sortino ratio). The benchmark compares proprietary (e.g., GPT-5, Claude-4) and open-weight (e.g., Qwen3, Kimi-K2, GLM-4.5) models.

**Strengths:**

1. The study addresses an interesting real-world challenge by evaluating AI agents for stock trading simulation.
2. The selected period in 2025 is thoughtful and helps mitigate the issue of potential data leakage.
3. The paper is clearly written and easy to follow, with well-organized presentation.

**Weaknesses:**

1. It would be helpful to clarify how the proposed AI agents compare to existing stock-trading agents. Several prior studies have already explored the potential of LLMs or AI agents in this context, and therefore, I am not very convinced with the novelty of this work.
2. Stock trading is highly complex and often influenced by sudden or unpredictable events, such as the COVID-19 pandemic, which caused sharp market fluctuations. The paper does not appear to demonstrate that the proposed system can handle or adapt to such abrupt disruptions.
3. To avoid data leakage, the study uses multiple periods within 2025. However, I wonder if this relatively short time frame can be insufficient to capture long-term trends or validate the robustness of model predictions, especially given the volatility of certain stocks.

**Questions:**

1. Although the study presents STOCKBENCH as a benchmark for AI agents, it seems closer to a stock-trading framework. Could the authors clarify how external researchers can access or benchmark their own models using this resource?
2. Is the benchmark primarily intended for performance measurement (evaluating trading outcomes) or for agent development (testing new architectures and strategies)?
3. Have the authors considered including comparisons with existing AI-based stock trading agents or traditional algorithmic approaches to better contextualize the benchmark results?

---

### Official Review · Reviewer_FKzD · 2025-11-01

**Soundness:** 3
**Presentation:** 3
**Contribution:** 2
**Rating:** 2
**Confidence:** 4

**Summary:**

This paper introduces STOCKBENCH, a benchmark for evaluating the ability of LLM agents to conduct stock trading.

**Strengths:**

The paper introduce a concept "contamination-free", which offers a new perspective for evaluating LLMs in temporal domains. The authors correctly argue that testing on past market data risks evaluating memorization, not reasoning.

**Weaknesses:**

1. Short evaluation period. The main evaluation period is only four months long, during which the buy-and-hold baseline was nearly flat (0.4% return). This makes it an easy benchmark to beat. The paper's own analysis (Sec 4.4) shows that in a different (downturn) period, all agents failed to beat the baseline. This suggests the main claim of "profitability" is highly dependent on the chosen market regime and not generalizable.

2. The justification for the portfolio construction is weak. The authors select the "20 stocks from the Dow Jones Industrial Average with the highest weights" but do not explain why this specific method was chosen over a random sample or a sector-stratified sample. This introduces a significant selection bias, and the agents' performance may not be generalizable beyond this specific 20-stock portfolio.

3. There is a major contradiction between the Abstract and the Main Results.
Abstract: "...most LLM agents struggle to outperform the simple buy-and-hold baseline..."
Section 3.2 Results: "Most tested models outperform the passive buy-and-hold baseline..."

4. The paper's "contamination-free" claim focuses on using the recent data, but it ignores a more-subtle and important form of contamination: memorized behavioral priors. The 20 chosen assets are the most-analyzed stocks in the world (top DJIA components). It's highly probable that the LLMs have memorized the typical behaviors, seasonal patterns, and characteristic responses of these specific companies from their training data. For example, the model might "know" from reading millions of historical analyses that "Stock X is highly sensitive to oil prices" or "Stock Y tends to beat earnings expectations and rise." This is also a form of data leakage that using recent data does not solve. Because this point is the only noverty comparing with the existing benchamark, you may want to design more careful design for this.

**Questions:**

Have you considered testing the agents on a 'shadow portfolio' of 'twin stocks'—i.e., closely correlated competitors from the same sectors as your 20 chosen stocks (e.g., Bank of America instead of JPM, or Oracle instead of Salesforce)? Running the simulation on this shadow portfolio would reveal if the agents learned generalizable, sector-wide strategies or if their performance is merely an artifact of the specific 20 stocks selected.

---

### Official Review · Reviewer_YVFQ · 2025-11-01

**Soundness:** 2
**Presentation:** 3
**Contribution:** 2
**Rating:** 2
**Confidence:** 4

**Summary:**

This paper provides a high-fidelity simulation benchmark for LLM to evaluate their decision making ability in stock market. Agents use historical data from a past period and simulated to prevent future information leakage

**Strengths:**

this simulated environment addresses the data contamination issue

**Weaknesses:**

The novelty is quite limited. There are already plenty works on evaluating LLMs' ability on stock market such as InvestorBench, StockAgent, Agent Market Arena.
In addition, a better simulation may just be evaluating LLMs on current marketplace, which is indeed feasible and completely remove the issue of data leakage. I wonder what is the reason that such an environment is not built

**Questions:**

1. Why only choose top 20 DJIA stocks?
2. The paper admits LLMs fail in bearish markets and that evaluation period matters hugely, but still draws conclusions from a single 4-month window?

---

### Note · Authors · 2026-01-04

I have read and agree with the venue's withdrawal policy on behalf of myself and my co-authors.